# Robust Detection for Chipless RFID Tags Based on Compact Printable Alphabets

**DOI:** 10.3390/s19214785

**Published:** 2019-11-03

**Authors:** Hatem Rmili, Boularess Oussama, Jawad Yousaf, Bandar Hakim, Raj Mittra, Taoufik Aguili, Smail Tedjini

**Affiliations:** 1Electrical and Computer Engineering, Department, Faculty of Engineering, King Abdulaziz University, P.O. Box 80204, Jeddah 21589, Saudi Arabia; bmhakim@kau.edu.sa (B.H.); rajmittra@ieee.org (R.M.); 2Communications Systems Laboratory, National Engineering School of Tunis, University of Tunis El Manar, BP 37, Belvedere 1002 Tunis, Tunisia; boularesoussama@hotmail.com (B.O.); taoufik.aguili@enit.rnu.tn (T.A.); 3Department of Electrical and Computer Engineering, Abu Dhabi University, 1970 Abu Dhabi, UAE; hjawadyousaf@gmail.com; 4EMC Lab, Electrical and Computer Engineering Department, University of Central Florida, Orlando, FL 32816, USA; 5LCIS, University Grenoble-Alpes, 26902 Valence, France; smail.tedjini@lcis.grenoble-inp.fr

**Keywords:** radio frequency identification (RFID), chipless RFID, radar cross-section (RCS), alphabet letters, resonant letters

## Abstract

This work presents a novel technique for designing chipless radio frequency identification (RFID) tags which, unlike the traditional tags with complex geometries, are both compact and printable. The tags themselves are alphabets, which offers the advantage of efficient visual recognition of the transmitted data in real-time via radio frequency (RF) waves. In this study, the alphabets (e.g., a, b and c) are realized by using copper etching on a thin dielectric substrate (TLX-8) backed by a ground plane. It is shown that the original signature of the frequency response of the backscattered radar cross-section (RCS) of the letter, displays dips that are unique to the individual letters. The tags have been simulated, fabricated and their monostatic cross-sections have been measured by using a dual-polarized Vivaldi antenna in the frequency band ranging from 6 to 13 GHz. The study also includes, for the first time, a detailed analysis of the impact of changing the shape of the tag owing to variation in the font type, size, spacing, and orientation. The proposed letters of the alphabet are easily printable on the tag and provide an efficient way to visually recognized them and, hence, to detect them in a robust way, even with a low coding density of 2.63 bit/cm^2^. The advantages of the proposed novel identification method, i.e., utilization of the both co- and cross-polar RCS characteristics for the printable clipless RFID tags are the enhancement of the coding density, security and better detection of the alphabet tags with different fonts by capturing the tag characteristics with better signal to noise ratio (SNR). Good agreement has been achieved between the measured and simulated results for both co- and cross-polarized cases.

## 1. Introduction

Chipless tags for a radio frequency identification (RFID) technology has drawn considerable recent attention, owing to its associated advantages of low cost as well as small size, in comparison to the traditional chipped RFID tags printed on silicon chips [1,2,3,4]. It is anticipated that chipless-RFID tags would also replace the barcodes in the near future, since they offer the additional advantages of no-line-of-sight requirement, enhanced encoding characteristics and reduced manufacturing cost, as compared to the barcode-based designs [5]. Low-cost printing advantage, offered by the chipless RFIDs, makes them a powerful candidate for numerous smart applications in the Internet of Things (IoT) [2,5], medical diagnosis [6], classified document security [1,7], low-cost shipment tracking and warehouse inventory control [4,7,8,9]. 

Chipless RFID tags fall into three main categories: (i) tags based on the time domain-reflectrometery (TDR) method [10,11]; (ii) tags utilizing the frequency domain that depend on the signature resonance properties of the reflected signals from the tag-IDs in the frequency domain [1,2,3,12,13,14]; (iii) tags based on hybrid approaches [15,16,17].

Chipless RFID based on time domain technique, such as surface acoustic wave (SAW) phenomena, are frequently used for some applications [10]. However, they are not fully printable and they suffer from the drawbacks of having a large size needed to integrate ultra-wide-band active and passive components on the tag, which in turn requires an A/D converter with a high sampling rate at the reader end [1,8,10,11]. Since the image-based chipless tags, operating on the principle of the Synthetic Aperture Radar (SAR) require the scanning of high-resolution images, the inherent associated cost of scanning hardware as well as the requirements of high computational image processing resources limit their practical applications [1,12,18] in which cost-saving is an essential issue.

Frequency domain tags based on resonant structures, offer the advantages of simplicity of design of both the tag and the reader structure, as well as higher coding density [1,2,3,4,7,8,13,14,19,20], as compared to their counterparts [1,8,10,11]. The data encoding procedure of a frequency domain tag is based on the encoding of its signature resonance state, namely peaks and valleys of the magnitude of the backscattered signal magnitude [8,13], phase [7,8,13,19] and/or the characteristics of its radar cross-section (RCS) [1,16,21], associated with the metallic geometry of the resonant tag-ID.

A wide variety of multipurpose chipless RFID tags and detection systems based on their frequency domain characteristics have been proposed in the literature. The examples include: space-filling curve tags [9]; stub-loaded structure [7]; multi-resonator without ground plane [17,22]; stacked multilayer patches [19]; slot-loaded patches [23]; barcode-based designs [12,18] and multi-resonator structures such as spiral resonators [8,13]; shorted dipoles [24]; and L-shape resonators [4,5,6].

Recently, the use of ‘text’ as a resonator has gained interest for smart IoT applications, document security, and other RFID applications [1,2,14,21,25,26,27]. Besides fulfilling the demands of hi-tech innovative advertisements, this approach has the advantages that the printed name of the product, company name or logo, and document identification (ID), etc., can be used as RFID tags [1,25,26], without having to separately install relatively costly, and complex resonant structures as proposed previously [4,6,7,8,9,12,13,19,22,24]. The manufacturing and maintenance cost factors become more important when installation of a separate RFID resonator is required on a large-scale shipment tracking product, an industrial inventory-control warehouse system, smart IoT devices, and/or on millions of documents (currency, cheques, etc.) that need to be secured. Additionally, a comparison of the RF response signatures of the tag-ID letters, with the database of their resonant properties [1,2] provides more reliable detection as compared to that of the traditional systems [4,6,7,8,9,12,13,19,22,24], where either a slight change in the complex interaction setup (for example, presence of nearby metallic objects) or the tag-geometry alter the resonance characteristics of the backscattered signals, and thus might lead to an inaccurate detection, since the above tags [4,6,7,8,9,12,13,19,22,24] cannot be visually recognized.

As for the text-based chipless RFID tags, Keskilammi et al. [25] have proposed joining the text letters in the form of a meander line dipole antenna for UHF-band RFID applications. Subsequently, several authors [2,14,21,26,27] have demonstrated that the metallic letter-tags could be used for various RFID applications by exploiting the resonance characteristics of the printed alphabets, which may be Latin [14,21,26,27], or Arabic [2]. Recently, Arjommandi et al. [1] have proposed a document verification scheme which only utilizes the cross-polar resonant RCS properties of the peyote symbols in the frequency range of 60 GHz, and cloud-based deep learning pattern recognition system for the recognition of the tag-IDs. Arjommandi et al. [1] have collected the frequency domain data for different positions of the tag-reader system (hybrid approach) by using a complex data gathering hardware system before sending it to a cloud system for post-processing. Additionally, it should be pointed out that Boularess et al. study [2] has only worked for single Arabic alphabets, while Singh et al. work [14,27] is only limited for single Latin alphabets. It has been shown by Demir et al. [21] that an introduction of open slits in the Latin alphabets can enhance the distinguishability of closely-spaced letters that exhibit a relatively flat response.

This study develops a novel design of low profile chipless alphabet-tags based on three resonant metallic letters. Note that the letter-tags are being used for the first time in this work to combine the classical visual recognition with their electromagnetic signature identification. Initially, an easy design with Arial font of size 15 mm was used to realize the tag, which is printed on a TLX-8 (relative permittivity: 2.55 and loss tangent: 0.0019) substrate with a bottom ground plane.

The coding density of the procedure proposed herein, is enhanced by using both the co- and cross-polar RCS characteristics of the backscattered signals, and by including the effects of geometrical variations. This is done in the process of encoding the data by employing frequency shift technique, which differs from approaches described previously [1,2,14,21,25,27]. The improved coding capacity (minimum 2.63 bit/cm^2^ with used three-letter words), achieved by using the proposed technique is very favorable when compared to several existing legacy alphabets-based designs [1,2,14,21,25,27]. This feature as well as the reconfigurable nature of the proposed design, make the proposed tag an attractive choice for various smart IoT applications such as smart stores, secret document security, efficient handling of large-scale shipment tracking and industrial warehouse inventory-control systems. As mentioned earlier, the product name, etc. can be printed on the tag by using metallic letters which limits the need for installation of separate chipless-tag as proposed elsewhere [4,6,7,8,9,12,13,19,22,24].

The rest of the paper is organized as follows: Section 2 presents the design of the proposed chipless tag alphabet letter and words. The details of the principle of the working method of Figure 1 are given in Section 3. Section 4 describes the simulation results for both co- and cross-polarization cases for the letter type of tag-IDs. A comprehensive and detailed analysis of the impact of the geometrical variations on the resonance characteristics of the tag-ID is presented in Section 5. Section 6 describes the developed methodology for encoding the tag data. Section 7 presents experimental measurements and practical results of real tag letters. Finally, some conclusions are drawn in Section 8.

## 2. Design of Chipless Tag Alphabet Letters

This section details the design of the English alphabet-tag used for the actual analysis by using full-wave numerical modeling and demonstrates the use of the proposed method in Figure 1. The alphabet letters can be designed for any font size and inter-letter spacing. In this study, the use of the topology for the realization of the alphabet letters (e.g., a, b and c) is based on the ‘Arial font’ with a font size of 15 mm and no click spacing between the neighboring letters. However, it should be pointed out that the analysis could be conducted for any type of letter, font, and size by using the proposed methodology.

The utilization of the ground plane under the metallic tag helps to enhance the selectivity of the resonance peaks by improving the quality factor of the resonator, as compared to the earlier legacy designs reported without the ground plane [1,2,14,21,25,27]. Next, an RFID reader is used to launch a dual-polarized plane wave to excite the tag as an interrogating signal. The process of data recognition for this tag is based on the generation of a particular electromagnetic signature in the backscattered signal, realized to the monostatic RCS of the tag, for both co- and cross-polarizations, in the frequency range of 6–13 GHz. As shown in Figure 1, a robust recognition of chipless alphabet tag is conceived out by using a 3-bit encoding of the recorded dual-polarized signature EM properties of the letter-tag to detect it accurately. To evaluate the effect of the changes in the geometrical properties of the tags on the robustness of its detection, we have carried out an extensive detailed analysis of the change in resonance troughs of the RCS signals with the changes in the structural properties of the letters fonts, e.g., the type, the size, the spacing and the orientation. To the best of our knowledge, this is the first time such results have been presented in the literature.

For the realization of the tag-IDs, the letters are first drawn by using the ‘text option’ in the 3D modeling software such as AutoCAD 2015 or 3DS Max. After that, the letter layout in imported to the CST Microwave Studio (MWS) simulator for further electromagnetic analysis. The imported letters are placed on a 0.5 mm thick TLX-8 substrate (ε_r_ = 2.55 and tan δ = 0.0019) backed by a ground plane whose size is 4 cm × 2 cm. The size of the substrate ground plane is adjusted to obtain the RCS resonance peaks of the backscattered signals from the letters in the frequency range of 6–13 GHz. The RCS analysis approach is preferred for the letter type of tag-IDs since their relative RCS is much smaller than that of the background; and, hence, it provides more sensitivity in actual measurements to accurately detect these tags [1,20,21,27].

Figure 2 details the dimensions of the designed tag ‘abc’. The three-dimensional (3D) exploded view of the designed tag is presented in Figure 3, which shows that the letter-tag constitutes a radiating copper layer printed on the top of the TLX-8 substrate, and that the ground plane is at the bottom of the 0.5 mm thin substrate. The basic idea of introducing the ground plane below the metallic tags is to enhance the selectivity of the resonance peaks due to the interference between the metallic letters and the ground plane. The presence of the ground plane under the metallic tag also increases the sensitivity of the identifying tag with good isolation between the nearby object and the tag, which is attributable to the enhanced quality factor of the resonators as compared to tags without ground plane [1,2,21,22,26,27]. The presented tag in Figure 2 and Figure 3 is the combination of alphabets ‘a’, ‘b’ and ‘c’. Similarly, any combination of single or multiple-letters-tags can be realized to form words.

## 3. Working Principle

### 3.1. Working Principle of Proposed Chipless Letters-ID

The basic operational chipped RFID system requires the placement of a separate antenna in the reader and in the tag. Subsequently, the chipped-tag is detected through the communication between the antennae installed on both the transmitter (reader) and the tag.

Since the chipless tags are passive, and since they are metal-only layered structure, their accurate detection makes the tag-reader design more complex, when compared to that of a simple reading system of chipped tags [1]. With the advancements in the coding density techniques, researchers have been able to reduce the size of the tag, and have been able to use the coded information in the recorded RCS at the reader antenna terminal for the detection of the tag-ID [1,2,3,14]. In this method, the resonant peaks are extracted from the electromagnetic footprint of the received signal spectrum originating from the resonating target tag.

The working principle of the proposed robust chipless RFID letter-tag detection is illustrated in Figure 1. In this design, the RFID reader consists of two dual-polarized wideband antennae, referred to as the transmit (Tx) and receive (Rx) antennae. The ‘Tx’ antenna excites the chipless metallic tag with an ‘interrogating dual-polarized plane wave signal’ while the ‘Rx’ antenna records the backscattered dual-polarized RCS signal reflected from the tag. The recorded RCS by the ‘Rx’ antenna contains the information about the unique electromagnetic signature of the corresponding tag.

Taking into account the form of letters, we have usually frequencies which are generated with both polarizations. When we have the co-pole, in fact we filter the response of the tag to keep only one polarization, whereas in cross polar, the effect of filtering is not present and we obtain all the resonances. In fact, with the co-pole, what we measure is the superposition of the exiting signal and the response of the tag, which is more sensitive and low response levels are observed. The measured level of the signal in co-pole measurement is usually higher in amplitude because we observe the superposition of the exciting signal with the tag response. In cross-pole, we separate the channels of excitation and response, which means that we measure signals only generated by the tag (with level widely low). We have the better SNR in this case.

The cross polarization method is valuable for all letters since the shape of the letter permit the appearance of vertical and horizontal currents, which is true for all letters except perhaps for the letter “I” or “l”. For these two last cases, the width is too much to be detected in the same band as for the “vertical” resonances (in fact the horizontal frequencies well exist, but are shifted toward higher frequencies (more than one hundred GHz)).

In this work, the recorded co- and cross-polar responses are designated as VV, HH, VH, and HV notations, respectively. The co-polarized VV RCS response refers to the case when the letter-tag is excited by a vertical plane wave (‘Vt’) from the ‘Tx’ antenna and the reader (‘Rx’) antenna analyzes the vertically received (‘Vr’) frequency encoded reflected signal from the tag, and vice versa for the HH response. Similarly, cross-polarized HV encoded response corresponds to the ‘Vt’ excitation of the alphabet tag-ID and analysis of the horizontally received (‘Hr’) backscattered signal at the receiver antenna terminal, and vice versa for the VH term (see Figure 4a) for pictorial illustration). The RFID reader antennae must have high port-to-port isolation for an accurate recording of the co- and cross-polar components of the reflected signals from the tags [3,4,19,24].

### 3.2. Resonant Metallic Letters

The main objective of this study is to design chipless alphabet-tags that are easy to fabricate and to propose a new robust technique for their fast identification keeping the constraints of their practical implementation in mind. For this purpose, the designed alphabet letters in Figure 3 appear to be a good choice as they can be easily recognized, either visually or through suggested RFID recognition method, as explained here.

Figure 3 shows that the designed tag corresponds to a resonant metallic letter, which is printed on a thin grounded dielectric slab. The interaction of the launched incident plane wave with the metallic tag introduces a surface current around the tag geometry. The intensity of the induced surface current on the tag depends on the specific resonance frequency corresponding to the structural properties of that tag. For the sake of illustration, the different simulated co- and cross-polar backscattered RCS signals for a single metallic letter-tag ‘a’ are shown in Figure 4b, which illustrates that the prominent resonances (dips) in the amplitude of co-polar (HH/VV) components of the backscattered RCS signal, and the corresponding peaks in its cross-polar (VH/HV) components, respectively. These are the signature resonance properties of this particular resonant ‘letter-a’ and are referred to as ‘*f*_HH_ (a)’ and ‘*f*_VV_ (a)’ for the co-polarization cases. The induced surface current around the letter-tag will be maximum at these resonance frequencies, as shown in Figure 4c,d at ‘*f*_HH_ (a)’ and ‘*f*_VV_ (a)’, respectively.

The results of Figure 4b demonstrate that the metallic geometry of a particular letter-tag with specific properties (geometry, font size, and font type, etc.) creates a signature reflection response in the backscattered RCS signal. It shows that the metallic letter actually behaves as an EM polarizer. The demonstrated diffraction principle of the metallic letters-tag in Figure 4b is used here to suggest an efficient data encoding algorithm for the chipless-tags.

If we place the three different metallic letters (e.g., a, b and c) on a dielectric substrate, then there would be three different independent signature resonance frequencies for each letter (more details in Section 4.1). Since the frequency signature of each letter-ID is different, the signature offers a great advantage for its fast and accurate detection by matching the frequency response characteristics to the database comprising of unique signature frequencies of each letter. In the proposed coding scheme, each bit-status would correspond to ‘1’ or ‘0’, respectively, depending on the presence or absence of the specific resonance frequency signature for the exposed letter-tag. The resonance response of the pre-determined system, without the metallic letter layer, can be measured for effective and accurate detection of the tag.

## 4. Simulation Method

The proposed letters-tag, designed as described above, are simulated by using CST MWS. Figure 5 shows the simulation setup for the realized chipless alphabet-tags. The tag-IDs under test are exposed to dual-polarized linear plane waves. The both co- and cross-polar backscattered signals from the letter-tags in the far field region are recorded by using both the horizontal (H) and vertical (V) electric field probes that are placed at *z* = 150 mm (*z* > 2D^2^/λ, where D is the maximum size of the tag and λ is the free-space wavelength at the lowest operating frequency) away from the tags under test. The in-between distance between the tags and the horizontal and vertical electric field probes is determined according to the far field condition because the chipless tags are used to communicate wirelessly from long distance. It also results in the minimum read range of the proposed method to be 150 mm. The simulation is performed for the frequency range of 6 to 13 GHz by using a frequency domain method and employing an open boundary condition in CST MWS.

### 4.1. Co-Polar Detection of Alphabet Tag-IDs

For the recording of the co-polar detection response of the letter type of tag-IDs, a vertical/horizontal plane wave is used to excite the letter-tag, and an identical vertical/horizontal polarized probe with the same polarity is placed at a distance of 150 mm away, to record the receiving co-polar backscatter RCS signals from the tags. The simulations are performed for different combinations of three alphabets ‘a’, ‘b’ and ‘c’. Figure 6, Figure 7 and Figure 8 show the recorded signatures of co-polar reflected RCS responses corresponding to each of the tag combinations (single alphabet, two alphabets, and three alphabets), respectively.

Figure 6 presents the co-polar (HH and VV) RCS responses for the single alphabet letter-tags ‘a’, ‘b’ and ‘c’. Figure 6a,b clearly show that that the resonant troughs in both co-polar VV and HH cases are different for each tag and that the resonance properties of the tags change as we change the incident and receiving characteristics of the plane waves. The observed resonance frequencies for tag-an in HH and VV polarization cases are 8.9 GHz (*f*_HH_ (a)) (see Figure 6a), and 10.97 GHz (*f*_VV_ (a)) (see Figure 6b), respectively. A similar observation can be made for the other tags b and c. These resonance troughs are formed by the interference between the letter-tag top metallic layer and the bottom ground plane. Furthermore, as expected, we note from Figure 6 that the resonance property of each letter-tag is unique, owing to the variance in the geometrical structure of each letter-tag which results in a different induced surface current distribution for each tag. Table 1 presents a summary of the corresponding unique signature frequencies (*f*_HH_ and *f*_VV_) for each letter-tag shown in Figure 6.

The co-polar spectrum responses of the other two realized tag configurations i.e., ‘tag-ac’ and ‘tag-abc’, are depicted in Figure 7 and Figure 8, respectively. We notice that there are two resonance dips in Figure 7 for the two letters ‘ac’, and three resonance troughs can be observed in Figure 8 for the three-letters tag. It demonstrates the fact that the unique resonance spectrum of each letter-tag is combined in the joined spectrum of the ‘tag-ac’ and ‘tag-abc’, respectively. Figure 7 and Figure 8 also demonstrate that the dips in the responses of the combined letters (‘ac’, and ‘abc’) are exactly at the same frequencies, as noted in the separate letter-tags ‘a’, ‘b’ and ‘c’, respectively. The effect of the presence and absence of the ground plane under the metallic tag structure is depicted in Figure 9 which shows that the presence of the ground plane results in clear resonance dips in both co- and cross-polarization cases. Thus, it makes the tags identification and differentiation process easy as compared to the RCS response without ground plane under the metallic tag structure. It demonstrates the relative robustness of the proposed procedure for accurate differentiation of the tags based on the designed resonant metallic letters with a ground plane when compared to previously designs [1,21].

The effect of the change in the resonance properties of the letter-tag with the variations in its position and its repetition is illustrated in Figure 10, which shows that the location modification or the repetition of the letters reproduce a maximum shift of 0.009 GHz which is approximately 0.1% of the total utilized spectrum bandwidth. It is a negligible variation and it has no effect on the unique electromagnetic signature of the letter-tags. The change of the location or repetition of the letter-tags varies the magnitude of resonance dips with minimum change in their unique resonance characteristics.

The intensity of the generated surface current around the alphabet-tag (for combined letters) varies according to the resonance properties of each letter-tag, for the exposed plane waves. The surface current distribution for the tag-abc, when excited with horizontal and vertical plane waves are shown in Figure 11 and Figure 12, respectively. These figures confirm that the ‘hotspot’ of the surface current magnitude either moves from top to bottom, or left to right, or vice versa, with the change in the resonance frequencies of the resonant letters. The spreading of the current in the conductive layer of the excited letter-tag can be observed in Figure 11 and Figure 12, and these current distributions are related to the observed resonance troughs in the frequency spectrum of the letters-tag in Figure 8.

Figure 11 shows the surface current distribution at the signature resonant frequencies of each corresponding tag (see Table 1) i.e., *f*_HH_ = 8.90 GHz for latter ‘a’, *f*_HH_ = 6.96 GHz for letter ‘b’, and *f*_HH_ = 8.32 GHz for letter ‘c’, for the case of horizontal plane wave excitation. Similarly, the current distributions at the corresponding co-polar frequencies for vertical plane wave excitation are shown in Figure 12. Indeed, by observing the current distribution results at *f*_HH_ = 8.90 GHz (see Figure 11a) and *f*_VV_ = 10.97 GHz (see Figure 12a), we can easily identify the ‘tag-a’, since the current levels are minimum for the other letters, namely b and c. Conversely, maximum current levels are observed at letter ‘a’ owing to its resonance properties at these frequencies. The current distribution density is maximum for letter b, while the minimum levels are observed for other two letters ‘a’ and ‘c’ at *f*_HH_ = 6.96 GHz (see Figure 11b) and *f*_VV_ = 6.97 GHz (see Figure 12b). The similar observation can be made for tag-c in Figure 11c and Figure 12c, which identify the tag-c for its unique resonance characteristics at the frequencies mentioned above.

The resonance frequencies *f*_HH_ and *f*_VV_ are almost equal when the letter ‘b’ is considered (see Table 1). It is because of the shape of the letter as the change of polarization did not bring any change in the resonance properties of the letter ‘b’ due to its structural circular shape which results in the almost identical current distribution and resonance frequencies in both horizontal and vertical polarization cases [28] as can be seen from the current distribution results of Figure 9 and Figure 11 for the letter ‘b’. According to Figure 11 and Figure 12 results, the induced current for latter ‘b’ is centered in the internal circle of the letter ‘b’ in both HH and VV polarization cases. Additionally, the level of induced surface current for letter ‘b’ is identical in both vertical and horizontal polarization cases. For this reason, the letter ‘b’ resonance frequency characteristics are almost insensitive to the polarization.

It can be inferred from the above analysis that a combination of the associated resonance frequency and the corresponding current distribution represent a unique EM signature for each letter-tag which can be used for its accurate detection as well as accurate decoding of the complete letter. Note that we can explain the reason for the absence or presence of the frequency signature of the identified tag-ID from the surface current density induced in the letter.

### 4.2. Cross-Polar Detection of Alphabet Tag-IDs

The orientations of transmitting and reading antennae play a vital role in the identification process of the chipless tags [1,2,20]. To study these effects, we have carried out the simulations of the tag-IDs described in the last section and have studied the cross-polar effects. For this study, we use a vertical plane wave to excite the tag (Tx. ‘V’) and record the backscattered RCS-HV response by using a horizontal probe (Rx. ‘H’). Likewise, a horizontal plane wave is used to illuminate the tag and a vertical probe (Rx. ‘V’) is used in the receiver to read the cross-polar RCS-VH response. The co-polar measurement probe needs to be rotated by 90° with respect to the plane wave polarity to measure the cross-polar results. The new position of the probe, as indicated in Figure 4a, reads the backscattered signals in cross-polarization. Therefore, we observe the interference peaks in the cross-polar responses instead of troughs that appear in co-polar results.

It is useful to note that both horizontal and vertical field components are generated in the cross-polar analysis. Figure 13 shows the simulated cross-polar RCS results for tag-a, -b, and -c, while Figure 14 presents the RCS results for tag-abc. We note that both cross-polar RCS-HV and the RCS-VH responses are almost identical for both separate and combined letters-tag cases. In each case, we compare the frequency responses obtained in cross- and co-polar cases. We note, by comparing the results in Figure 6 (co-polar) and Figure 13 (cross-polar) that for the cross-polar scenario, we have all the frequency signatures (interference peaks) as observed in the resonance troughs of co-polar HH and co-polar VV cases in Figure 6. Additionally, a similar trend can be observed by comparing Figure 8 and Figure 14.

Since the backscattered encoded frequency signatures are independent of tag orientation, these designed tags ensure robust detection owing to their ability to identify the tags regardless of the polarization. Recognizing the tag by using the dual-polarized data also facilitates the process of encoding of the letter-IDs whose resonance characteristics become similar for one of the polarizations [1,21,25] which, in turn, enhances the overall performance efficiency of the proposed robust detection, irrespective of the positions of the alphabet-tags.

## 5. Impact of Tag Modification on the Resonance Properties of the Tag

It is useful to understand the changes in the background RCS signal characteristics when we change the position of the tag, or we use different styles for writing alphabets such as different spacing between letters, font style, and the size of the font being used to write the characters. The sensing mechanism of the tag-ID relies upon the effect on its resonance frequency as its configuration is modified.

### 5.1. Impact of 45° Polarization on the RCS Response

This section elaborates on the impact of the change in the excitation position of the tag-IDs on its RCS response. It is possible to launch the two different polarization signals to excite the tag-IDs concurrently for the recording of the backscattered responses simultaneously by using different probes. Moreover, the orientation of the excitation and reading antennae are important key parameters which determine the accuracy of the identification.

To study these effects, we have simulated all the alphabets with an orientation angle of 45°. In this situation, both the horizontal and vertical field components are generated simultaneously when exciting the tag-IDs. In each case, we compare the resonance frequencies obtained for the three orientations of the excitation: horizontal, vertical, and 45°. Figure 15 shows the results for the two tags: tag-an and tag-abc, respectively. We notice from this figure that for 45° orientations, we have all the signature frequencies for horizontal and vertical cases. We observed that the results for the cross-polar cases reported previously, were essentially identical to the polar 45° results presented in Figure 15. This leads us to conclude that a robust system composed of just two antennae could identify the tag even when it is oriented along 45°.

### 5.2. Effect of the Space Between the Letters on the RCS Response

We now analyze the effect of the spacing between the letters in the realized tag-abc by investigating different simulated tag-models in Figure 16a. All nine resonant frequencies representing the tag-ID of ‘abc’, ‘a bc’ and ‘a b c’ can be observed in the simulated co-polar RCS results in Figure 16b. It shows that a slight variation in the spacing between the letters does not create problems with the identification of the tag, printed with or without the extra spaces in-between the letters. This is because the difference between the shapes of the resonance troughs is almost negligible for different cases shown in Figure 16a, owning to the installation of the ground plane under the printed tag-IDs. Consequently, we can use this technique of separation between the letters to increase the number of coding combinations introduced by this type of tag.

### 5.3. Impact of the Letters Font Type on the RCS Response

The readability of these typographic alphabets is maintained as we vary the font type. Figure 17b presents a comparison between the simulated RCS-HH spectra corresponding to the printed letters that have the standardized fonts such as Arial, Corbel and Times New Roman. Figure 18 compares the variations in the resonance properties of the tag-abc with the variations in the writing style (gras or regular) of the same type of font, e.g., Arial. We note from Figure 17b that the resonance properties of the tags have minor variations with the changes in the type of the fonts. Figure 17b and Figure 18b demonstrate that a change in the font type can introduce a frequency shift which ranges from 95 MHz (maximum) to 8MHz (minimum). These frequencies shift characteristics could be used to enhance the coding capacity of the proposed detection system as explained in Section 6.

### 5.4. Impact of Font Size (Height) of the Letter on Its RCS Responses

To study the impact of the size of a letter font on its RCS characteristics, we perform a parametric sweep in which the font style is fixed as Arial, while the heights of the letters for the tag-abc are varied from 16 mm to 16.4 mm in a 0.2 mm coarse step size. Figure 19a depicts the simulated models of the tags with different sizes. The backscattered co-polar HH frequency response is simulated and the RCS results are depicted in Figure 19b. Figure 19b curves show that a change in the height of the letter by 0.2 mm will shift the resonance frequency by 7 MHz for each letter. Figure 19b shows that, in general, a change of the font size will shift the spectrum signature without affecting its novel ID signature corresponding to the alphabet-tag.

We have analyzed the effect of modifying the geometrical parameters of the letters, namely the orientation, type of the font, the inter-character spacing, and the font height (size) for the fixed TLX-8 dielectric substrate and ground size, as mentioned in earlier Section 2. Figure 15, Figure 16, Figure 17, Figure 18 and Figure 19 show that different RCS signatures of the letters are different for different font parameters. Consequently, the coding density of the tag can be increased by considering all of these factors.

## 6. Developed Encoding Method for Chipless Tag-IDs and Coding Density

### 6.1. Encoding of Chipless Alphabet-Based Tags

One of the main encountered challenges during the development of a chipless RFID tag is the corresponding development of the digital encoding technique. The conventional techniques used for the encoding of the chipless RFID tags are based on the presence or absence of a peak or trough in the recorded spectrum. The presence and absence of a resonant frequency is used to encode logic-1 and logic-0, respectively. Additionally, unlike legacy designs [3,7,8,9,13,15,19,20,24], employing this method of tag-detection based on the presence or absence of a resonant frequency in the spectrum can also be used to encode the data along with visual analysis technique that corresponds to the presence or absence of the resonant frequency or letter-tag. The ID signature encodes the tag-abc as ‘111’, since there are three letters present. The logic code can be modified either by eliminating the corresponding letter, or by changing its geometrical parameters as mentioned in the previous Section 5. For example, a tag-ID of ‘101’ can be obtained by eliminating the letter ‘b’, which is at the second position.

The change of the ID-bit from ‘1’ to ‘0’ is done by marking the absence of the resonant trough in the tag-ID EM signature. Figure 8 presents the simulation results of the original ‘tag-abc’, written with an Arial font of 15 mm size. The RCS characteristics of four other reconstructions of this original tag-abc are illustrated in Figure 6 and Figure 7, respectively. Figure 6 depicts tag-a, -b, and -c, while Figure 7 shows the reconstructed configuration with the elimination of the letter ‘b’, i.e., tag-ac. We observe three troughs that were labeled as ‘111’ in the original tag-abc simulation results of Figure 8, and the corresponding resonant frequency points are 8.90 GHz, 6.96 GHz, and 8.32 GHz in the co-polar HH, and 10.97 GHz, 6.97 GHz and 12.44 GHz in the co-polar VV results, respectively. The tag-ID can be changed from ‘111’ to ‘101’, which corresponds to tag-ac. When the b-shape is removed, the resonance frequency points shift to 8.90 GHz and 8.32 GHz for the case of co-polar HH, and to 10.97 GHz and 12.44 GHz for the case of co-polar VV cases, respectively. Finally, one resonance dip, denoted by ‘100’, ‘010’ and ‘001’ in each reconstruction of the original tag-abc, corresponds to tag-a, -b, and -c, respectively. Unlike the traditional identification techniques for chipless tags [3,7,8,9,13,15,19,20,24], the proposed technique changes the encoding-ID from ‘1’ to ‘0’ by removing the alphabet(s) from the letter-tag. Additionally, it shows that the chipless alphabet-based tag-IDs can be optically controlled.

### 6.2. Shift Frequency Method and Coding Density

The coding strategy for the chipless RFID tags is dependent on the employed method of coding such as frequency shift coding [1,20,21,29], and phase deviation coding [7,8,13,19]. The findings of these studies demonstrate that the resonance frequencies of the chipless alphabet-based printed tags are strongly linked to their structural properties. It is evident that one resonant letter-tag can encode many bits instead of just one [1,20,21]. This leads to a substantial improvement in the coding capacity of the chipless tags and enhances the configurability of the coding method as well.

In this work, we use the resonance frequency shift as a parameter to identify the printed tag-abc, which has been realized in different geometrical forms (changes of font style, size and letter-spacing) as explained previously in Section 5. The normalized magnitude of the percentage resonance frequency shift ($\Delta f_{r}^{}$) can be expressed as:(1)$$\Delta f_{r}(\%)=\frac{f_{r}^{\mathrm{mod}\_\mathrm{tag}}-f_{r}^{\mathrm{org}\_\mathrm{tag}}}{f_{r}^{\mathrm{org}\_\mathrm{tag}}}\times 100$$

In Equation (1), $f_{r}^{\mathrm{org}\_\mathrm{tag}}$ is the original resonance frequency of the tag-abc, designed with an Arial font, a fixed height of 15 mm with the letters printed without a space in-between. Here, $f_{r}^{\mathrm{mod}\_\mathrm{tag}}$ is the modified resonance frequency of the tag-abc following the changes in the letters fonts, the style, the size, the orientation and the in-character spacing. The numerical sign of $\Delta f_{r}^{}$ shows the percentage of the shift in the resonance frequency of the modified tag-abc, as compared to the resonance frequency of the original tag-abc. These results are subsequently used to characterize the recognized tag-abc.

Examining the simulated RCS-HH response of tag-abc printed by using the various fonts in Figure 17, we observe that the tag-abc with Time and Corbel fonts linearly shift the original resonance frequencies, i.e., $f_{r-HH}^{\mathrm{org}\_\mathrm{tag}}$ (a,b,c) corresponding to the tag-abc with original Arial font (See Table 1)) to both lower and higher frequency ranges. A comparison of the sensitivity analysis results of Figure 16, Figure 17 and Figure 19, with those of the original configuration in Figure 8, obtained by using Equation (1), is presented in Table 2.

It is evident from Table 2 that the modification of the alphabets ‘a’ and ‘c’ font to ‘Times font’ will result in approximately 4.49% or 0.4 MHz ($\Delta f_{r-HH}^{}$ (‘a’)) and 6.00% or 0.5 MHz ($\Delta f_{r-HH}^{}$ (‘c’)) shift in the detection sensitivities of these above letters, based on their co-polar HH responses, respectively. On the contrary, the modified letter ‘b’ shifts the resonance frequency toward a lower frequency, e.g., by $\Delta f_{r-HH}^{}$ (‘b’) = −1.72% or −0.12 MHz. Similarly, a change in the frequency shifts for the other structural variation, the height and the spacing in the tag-abc can be observed in the results presented in Table 2. On the basic of the variations in the resonance frequencies of tag-abc presented in Table 2, we can justify saying that there exists a novel ID, provided that the resonance frequency shifts to another frequency with the variations in the tag structure.

Furthermore, the obtained results in Table 2 confirm the possibility for combining several geometrical parameters of the letter-tag in the encoding process as compared to the legacy designs which only rely upon the absence or presence of the EM resonance in the coding procedure [1,2,3,7,8,9,13,14,15,19,20,21,24,25,26,27]. Additionally, a larger number of combinations are possible in the encoding procedure when we change more than one parameter of the font for a given resonant letter, as for instance the size, or the style, and/or the spacing in-between the letters, as shown in Figure 16, Figure 17, Figure 18 and Figure 19, consequently, it enhances the coding density per surface area of the tag.

The algorithmic flow chart of the proposed method for the robust detection of the alphabet tags is depicted in Figure 20, which shows the steps of the developed algorithm for the robust and accurate detection of the tag-IDs based on the frequency shift analysis of their recorded co- and cross-polar RCS responses. The calculated percentage change ($\Delta f_{r}^{}$) in resonance frequency of the particular tag-ID configuration is compared with the pre-stored resonance frequency shift information about that specific tag-ID configuration. Lastly, the tag-ID is decoded based on the matching between the saved frequency shift value and the calculated frequency shift value. Besides the decision based on Figure 20 algorithm, the tag-IDs can also be optically detected which increases the robustness of the proposed method.

The frequency shift coding technique, employed in the encoding process in this work provides the possibility of defining more than one value of the resonance frequency, depending on the variations in the geometrical parameters of the letters-tag. For example, with four different values of each parameter of the font, it can be encoded to 5-bits for a single resonant letter, i.e., 32 possible combinations per one variation in the geometrical parameters of the font. To get an idea of the efficiency of the parameters of the changed font that we have introduced herein, the state variations in the three geometrical parameters (i.e., the font type, the size, and the spacing) of the letters are elaborated in Table 3. Based on this simple illustration, we can say that the tag can be encoded such that *P (geom_param)* represents 32 states, as may be seen by referring to Equation (2) below, by using only three geometric parameters. The number of combinations that can be encoded for each resonant letter by using the four parameters (i.e., *P (4 geo_p)*) is equal to 128, as may be seen from Equation (3) below. As a result, for the three resonant letters (‘abc’) used in our analysis, the total number of combinations is equal to 128^3^ = 2,097,152, as calculated in Equation (4) below, i.e., 21 bits within a tag the size of which is 4 cm × 2 cm. This corresponds to a minimum density of coding per surface (DPS) of 2.63 bits/cm^2^. It deduces that a large number of coding combinations can be generated with this new proposed coding technique based on the changes in the geometrical parameters of the letters combined with the frequency shift method.
(2)P(geo_p)=P(Fonts)∪P(size)∪P(spacing)=2∪4∪4=32
(3)$$\begin{array}{ll} P\left(4\;geo\_p\right) & =P\left(gep\_p\_1\right)\cap P\left(gep\_p\_2\right)\cap P\left(gep\_p\_3\right)\cap P\left(gep\_p\_4\right)\\ & =\left(2\cup 4\cup 4\right)\times 4=32\times 4=128 \end{array}$$
(4)$$P\left(letters\right)=P\left(letter\_a\right)\cup P\left(letter\_b\right)\cup P\left(letter\_c\right)=128^{3}=2097152\;\;\;\;\;\;\;\;\;\;\;\;\;\;$$

Table 4 lists a comparison of the proposed letters-tag performance with several existing legacy designs of chipless tags based on alphabets. The table illustrates that in fact the proposed RFID chipless tags based on resonant letters have a great potential for the high coding capacity, in comparison to all of the existing designs based on alphabets [1,2,14,21,25,27]. The higher coding density is achieved because the proposed encoding strategy is based on frequency shifting together with the incorporation of the impact of the structural variations in the writing styles of the tag-IDs. Furthermore, the most important characteristic of the proposed alphabet-based tags is the ability to optically control and detect all the resonances, without the ambiguity, in contrast to the traditional designs of chipless tag reported previously [4,6,7,8,9,12,13,19,22,24]. To add the security, feature to this letter-type of the tag-IDs, we can depend upon the modification of the geometric parameters. Moreover, if the writing style of the proposed letter-tags is increased, it would lead to an enhancement of the coding density from the minimum reported value of 2.63 bit/cm^2^ for the three-letter words. This gives a unique reconfigurable aspect to the proposed alphabet-tags when compared with that of the designs previously reported [2,14,21,25,27]. Although the reconfigurable feature is also present in a recently reported study [1], its coding density performance is inferior to that of the proposed design. Moreover, the above study does not include the effect of geometrical variations in the encoding process it follows. Overall, the observed coding density of the proposed design is superior to those described elsewhere [2,14,21,25,27]. Table 4 presents a comparison which confirms that the proposed design is well suited for the emerging smart applications, such as IoT, documents security, mass inventory handlings, as well as low-cost shipment tracking with an efficient robust detection of the printed alphabet-tags.

## 7. Experimental Verification

### 7.1. Realization of Letter-Tag IDs

To validate the new chipless tag designs, we have investigated the letter-tags corresponding to seven different combinations of alphabets ‘a’, ‘b’, and ‘c’ shown in Figure 19. The alphabet shapes were printed by using copper etching on a Taconic TLX-8 substrate, having a thickness of 0.5 mm. Each realized letter-tag of Figure 21 has a standard Arial font with a size of 15 mm, in order to operate in the ultra-wide-band frequency spectrum of 6 to 13 GHz with a multi-directional radiation pattern.

### 7.2. Measurement Setup

The simulation results discussed in the previous sections were verified with a monostatic measurement setup in an anechoic chamber. Figure 22 details the measurement strategy for the identification of the chipless alphabet-tags. The interrogation system of Figure 22 illustrates the actual. measurement setup (see Figure 23) for the recording of the two-port *S*-parameters by using a dual-polarized Vivaldi antenna. A dual-polarized Vivaldi antenna, with a transmit-receive gain of about 12 dBi in the frequency range of 0.7 to 18 GHz serves as an interrogation system (see Figure 22). It is connected to a two-port Agilent PNA-N5222A Vector Network Analyzer (VNA) (see Figure 23a). The VNA delivers the transmit power of 0 dBm to the attached Vivaldi antenna in the frequency range of 6 to 13 GHz. The chipless letter-tags are placed at a distance (*z*) of 15 cm from the reader antenna, as shown in Figure 23.

### 7.3. Recording of Measurement Data

A calibration technique mentioned by Hotte et al. [30] was used to nullify the background electromagnetic noise before measuring the co- and cross-polar backscattered fields of the letter-tag using VNA. Toward this end, we first performed a measurement without the chipless tag to characterize the environment and saved it as ‘Memory data’ in the VNA with the name of ‘No-tag S-parameter’ response. Subsequently, the normalized S-parameter responses for each letter-tag were measured and saved in the system to obtain the ‘normalized S-parameters tag’/ ‘S-parameter no-tag’. Finally, the calibrated response of the letter-tag was recalculated by subtracting the ‘S-parameters no-tag’ from the ‘normalized S-parameter tag’. Next, we measured the calibrated transmission and reflection coefficients, i.e., S_11_ and S_22_ corresponding to the co-polar RCS responses, and S_12_ and S_21_ representing the cross-polar characteristics. The two-port S-parameters of each realized letter-tag were measured using the procedure above described.

The measured two-port S-parameters for each tag-ID were further processed using MATLAB to obtain the co- and cross-polar RCS values of the tag-IDs (see Figure 22), from Equations (5) and (6), which are derived from the radar range equations [30].
(5)$$\sigma^{Co-polar}=\frac{R^{4}{\left(4\pi \right)}^{3}}{G_{t}G_{r}\lambda }\left(\left|S_{11/22}^{Co-tag}-S_{11/22}^{Co-no-tag}\right|\right)$$
(6)$$\sigma^{Cross-polar}=\frac{R^{4}{\left(4\pi \right)}^{3}}{G_{t}G_{r}\lambda }\left(\left|S_{12/21}^{Cross-tag}-S_{12/21}^{Cross-no-tag}\right|\right)$$

In Equations (5) and (6), *R* is the distance (*z*) between the tag being measured and the interrogation antenna; λ is the free space wavelength;$\;G_{t}$ is the transmit antenna gain; and $\;G_{r}\;$ is receiver antenna gain. The $\sigma^{Co-polar}$ and $\sigma^{Cross-polar}$ represent the computed co- and cross-polar RCS values from their corresponding backscattered *S*-parameter components, respectively. Additionally, the $S_{11/22}^{Co-tag}$ corresponds to the measured co-polar data received over the transmitted co-polar signal in the presence of the tag, and $S_{11/22}^{Co-no-tag}$ denotes the measured co-polar response received over the transmitted co-polar signal in the absence of the tag. For cross-polar RCS, $\;S_{12/21}^{Cross-tag}$ corresponds to the measured received co-polarized data over the transmitted cross-polarized signal in the presence of the tag and $S_{12/21}^{Cross-no-tag}$ depicts the measured received co-polarized characteristics over the transmitted cross-polarized signal in absence of the tag.

### 7.4. Measurement Results and Validation

In this section, we discuss the experimental validation of the simulation results presented earlier. The main objective is to study the backscattered RCS characteristics from the letter-tags in both the co- and cross-polarizations by using Equations (5) and (6), respectively.

To highlight the strategy followed for identification of each proposed alphabet-tag by using co-polar RCS results, the measured data for the same are plotted in Figure 24. Figure 24 shows clearly that the predicted resonance troughs in the simulations in all of the measured results for four different combinations of letter-IDs. The measurement results of the three-letter-tags ‘a’, ’b’ and ‘c’ are grouped in Figure 24a, while the results of ‘tag-abc’ are shown in Figure 24d. We note that the resonance dips are present in all of the three tags, i.e., ‘tag-a’, ‘tag-b’ and ‘tag-c’. Similarly, by examining the RCS results of ‘tag-a c’ presented in Figure 24b, we see that the resonant dips corresponding to the ‘tag-a’ and ‘tag-c’, are clearly distinguishable. The same observations can be made for the results of Figure 24c (for ‘tag-ab’) and Figure 24d (for ‘tag-abc’), where all the signature EM resonances corresponding to each individual alphabet can be seen quite clearly.

Figure 25 presents the cross-polar RCS measurement results by using Equation (6). We can observe that the resonance peaks appear instead of the troughs in the same signature frequency corresponding to each letter-tag ID. Additionally, in common with the simulation results, we obtain identical results for the measured RCS-HV and RCS-VH cases, as may be seen in Figure 25 for different tag-IDs. These results demonstrate that the depolarized chipless tags are suitable for the robust detection of printed data-ID.

Finally, a complementary lookup table was constructed by mapping the resonances to the letter-tag IDs for identification purposes. From the constructed table, one can identify and recognize the alphabet-tags simply by comparing the measured resonance frequencies of the respective letter-tags, for both polarizations.

Table 5 presents the measured frequency signatures, as well as the digital code corresponding to each letter-tag with Arial font of size 15 mm. A comparison of results in Table 1 with those in Table 5 confirms that the simulated and the measured signature resonance frequencies of alphabet-tags are in very good agreement for the different tags. Along with the measured signature resonance frequencies of the letter-tag IDs, Table 5 also presents their generated binary IDs. We note that there are eight possible encoded tag combinations in Table 5, which corresponds to a coding capacity of 3-bits. The advantage of the proposed approach is that each bit related to the presence or absence of the shape of the letter is written on top of the tag. For example, the binary code of the ‘tag-abc’ is 111 and by contrast, the binary code of the ‘tag-a c’ is 101. The constructed lookup table, for instance Table 5, can be used for any combinations of alphabets for an efficient and robust recognition of the alphabet-based tag-IDs.

In this study, we employed the RF detection system for the identification and recognition of the tag-IDs. The RF system have limitations of the more prone to the external electromagnetic (EM) noise and lesser read range as compared to the optical systems [31]. Optical system could also be used for the tags reading over the long distances. However, the optical system requires direct contact with the object to be identified. On the other hand, the RF system can even locate the objects at the sufficient long distances and even in the dark, which makes this method a favorable choice for different smart applications such as in IoT and cost-effective inventory management, etc.

The presence of external EM noise could bring a change in the resonance frequency and depth of the resonance value of the frequency signature of the tag-ID. For such cases different post data post-processing techniques like quantile regression models [32] and random sample consensus (RANSAC) algorithms [31] with other approaches such as anti-collision time domain techniques [33] and multi-antenna detection system [34] could be used to improve the detection efficiency and capabilities of the proposed robust RF-based detection system with the mitigation and suppression of the coupling noise.

The possible future work of the study could include the determination of the relationship between the ability to actually detect the tags and their visual recognition and development of the robust algorithms for the minimization of the external noise effect on the detection capabilities of the RFID detection system. Additionally, to ease the detection of the letters such as “i” and “I”, the designing of the special kind of typeface such as old Optical Character Recognition (OCT) and Magnetic Ink Character Recognition (MICR) fonts and their effect on the detection process could be a potential future research item.

## 8. Conclusions

In this study, the EM signatures of novel chipless RFID letter-tags was investigated experimentally by using three resonant alphabets (e.g., a, b, and c), to facilitate the design for different chipless RFID applications. The presented approach herein is an attempt to address the current challenges of efficient visual identification and recognition of the chipless tags by proposing the new alphabet-based tags. The standardized alphabets ‘a’, ‘b’, and ‘c’ with Arial font and height of 15 mm were realized by printing them on top of a thin-slab of Taconic TLX-8 substrate whose dimensions were 4 cm × 2 cm. Different combinations of letter-IDs were realized to demonstrate the proposed encoding procedure. The EM signatures for both horizontal and vertical field polarizations have been simulated in the frequency band of 6–13 GHz, and very good agreement between the resonance troughs of the simulated and fabricated tag-IDs have been realized. A comparison of the simulation and measured results shows that utilizing both the co- and cross-polar RCS responses enables one to accurately identify the letter-IDs owing to the unique frequency signatures associated with the tags. The detection of tags based on proposed both co-and cross-polar backscattered signature resonances did not only enhance the condign density but also allows the robust identification of the tags with better SNR and enhanced security. Along with the ability to carry out a robust detection, the proposed design of the chipless alphabet-based tag-IDs has several attractive features. These include the simplicity of reconfigurable design, low profile, substantial data capacity, improved coding density and orientation insensitivity. The proposed encoding procedure, which is based on frequency shift encoding, provides superior coding density of 2.63 bit/cm^2^ i.e., identification through the 21-bit encoding per surface area of 8 cm^2^ when compared to the reported legacy designs of alphabets-based tag-IDs. The efficiency of the proposed coding technique can be further enhanced, by increasing its coding density by incorporating the changes in the geometrical parameters (fonts) of the tag-IDs.

## Figures and Tables

**Figure 1 sensors-19-04785-f001:**
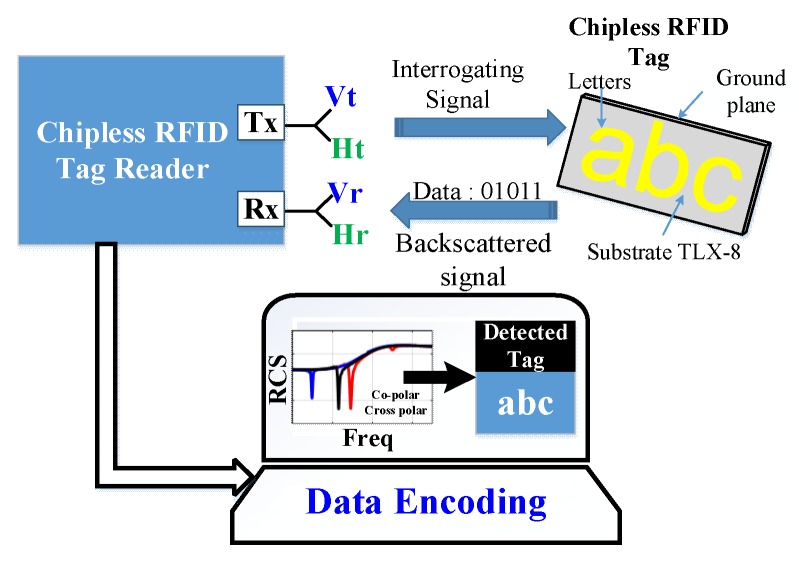
Working principle of the proposed methodology for the robust detection of the chipless RFID alphabet-tags.

**Figure 2 sensors-19-04785-f002:**
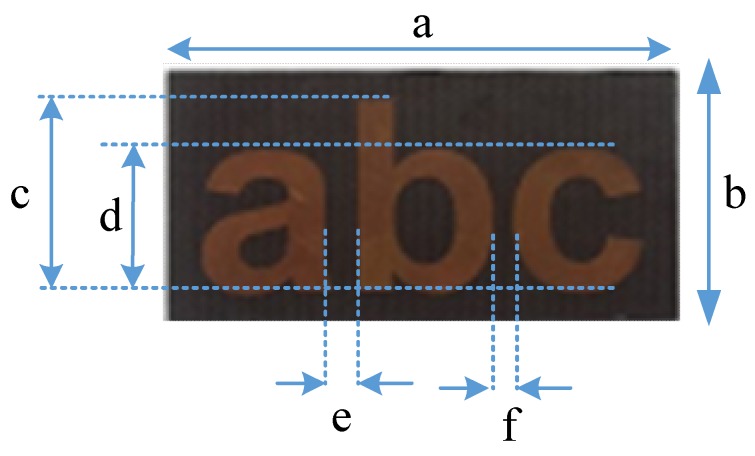
Top real view of the letter tag-abc. Values of the geometrical parameters are: Font = Arial, Font size = 15, a = 40 mm, b = 20 mm, c = 15 mm, d = 11.35 mm, e = 1.90 mm, and f = 1.5 mm.

**Figure 3 sensors-19-04785-f003:**
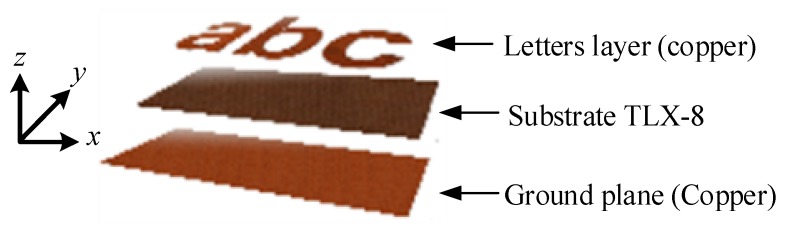
3D exploded real view of the letter tags showing the two metalized faces from each side of a common planar TLX-8 substrate of size of 4 cm × 2 cm.

**Figure 4 sensors-19-04785-f004:**
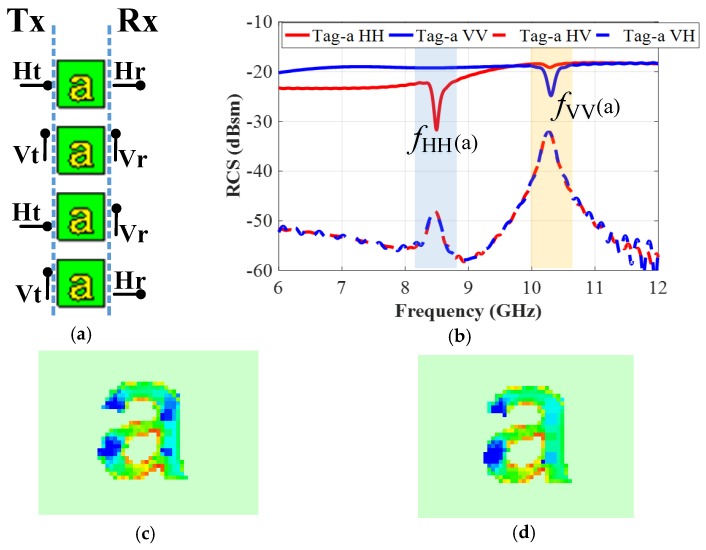
Simulation of the electromagnetic response of the “letter-a” resonator as a function of polarization: (**a**) Different polarizations setups (HH/VV/HV/VH); (**b**) Observed signature frequencies in co(HH/VV)- and cross(HV/VH)-polarizations; (**c**) Surface current at fHH (a) when tag is exposed to horizontally polarized plane wave (Ht) and probe record signals in both polarizations; (**d**) Surface current at fVV (**a**) when tag is exposed to vertically polarized plane wave (Vt) and probe record signals in in both polarizations.

**Figure 5 sensors-19-04785-f005:**
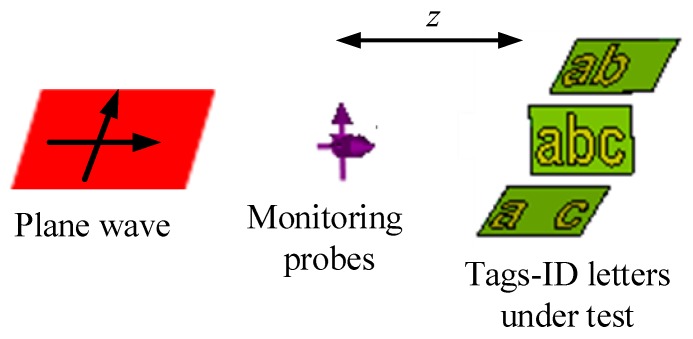
Simulation setup for the alphabet-based chipless tags.

**Figure 6 sensors-19-04785-f006:**
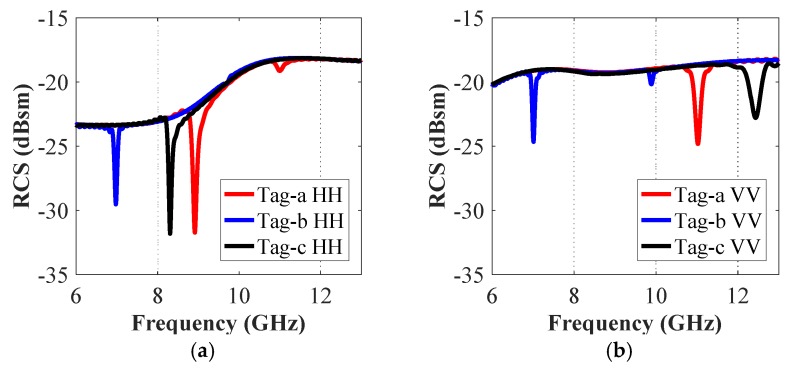
Co-polar RCS simulation results of tag-a, tag-b, and tag-c: (**a**) Co-polar HH RCS responses; (**b**) Co-polar VV RCS responses.

**Figure 7 sensors-19-04785-f007:**
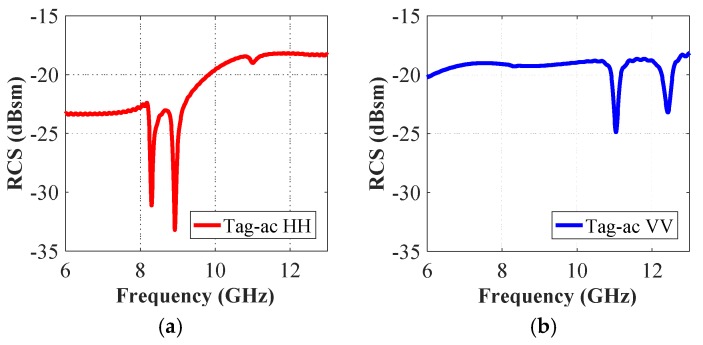
Co-polar RCS simulation results of tag-ac: (**a**) Co-polar HH RCS responses; (**b**) Co-polar VV RCS responses.

**Figure 8 sensors-19-04785-f008:**
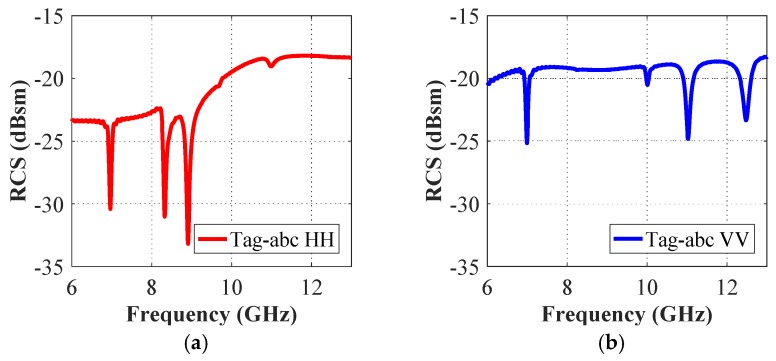
Co-polar RCS simulation results of tag-abc (**a**): Co-polar HH RCS responses; (**b**) Co-polar VV RCS responses.

**Figure 9 sensors-19-04785-f009:**
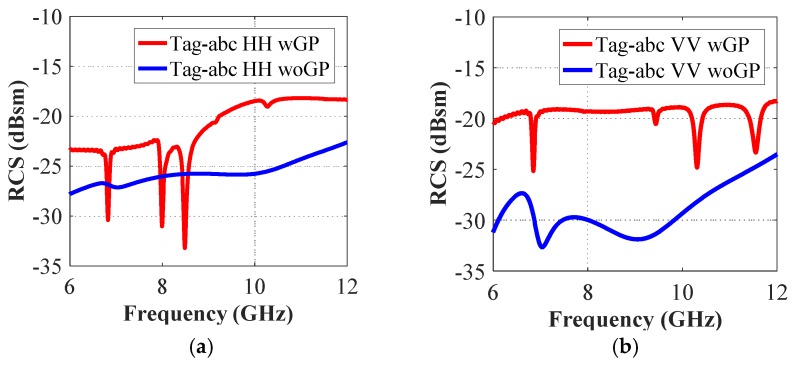
Co-polar RCS simulation results of tag-abc with and without the ground plane under the tag structure (**a**): Co-polar HH RCS responses; (**b**) Co-polar VV RCS responses.

**Figure 10 sensors-19-04785-f010:**
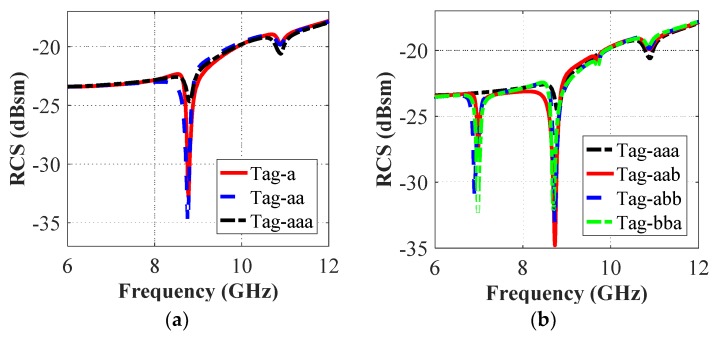
Cross-polar HH RCS simulation results (**a**) Letter place change effect; (**b**) Letter repetition effect.

**Figure 11 sensors-19-04785-f011:**
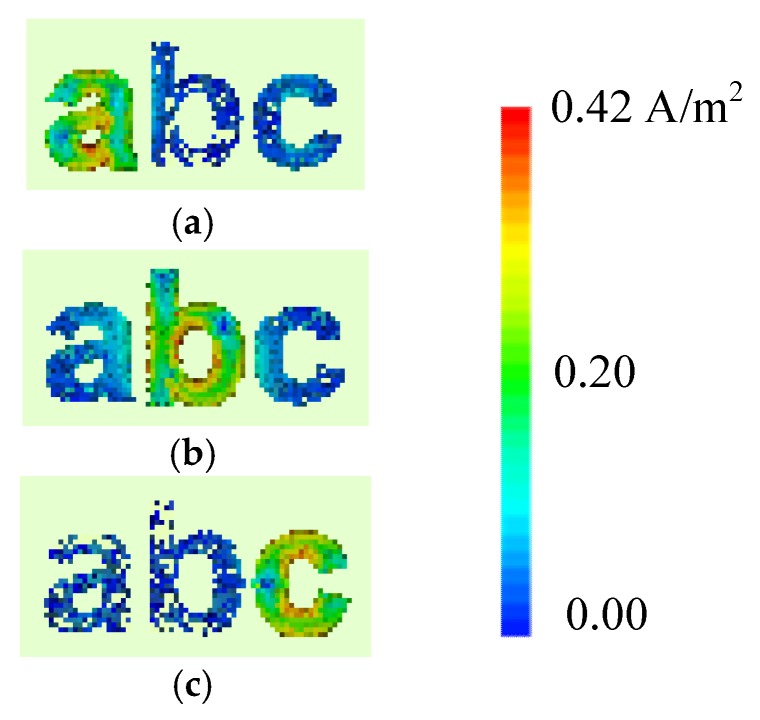
Surface currents obtained for the tag-abc excited with horizontal plane wave: (**a**) at unique frequency signature corresponding to letter ‘a’ i.e., *f*_HH_ = 8.90 GHz; (**b**) at frequency signature corresponding to letter ‘b’ i.e., *f*_HH_ = 6.96 GHz, and (**c**) at frequency signature corresponding to letter ‘c’ i.e., *f*_HH_ = 8.32 GHz.

**Figure 12 sensors-19-04785-f012:**
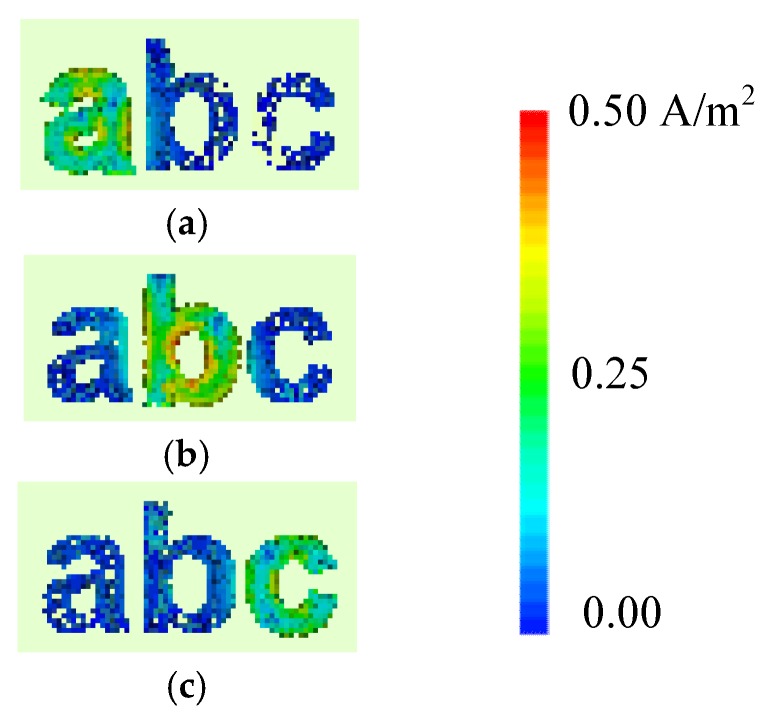
Surface currents obtained for the tag-abc excited with vertical plane wave: (**a**) at unique frequency signature corresponding to letter ‘a’ i.e., *f*_VV_ = 10.97 GHz; (**b**) at frequency signature corresponding to letter ’b’ i.e., *f*_VV_ = 6.97 GHz, and (**c**) at frequency signature corresponding to letter ‘c’ i.e., *f*_VV_ = 12.44 GHz.

**Figure 13 sensors-19-04785-f013:**
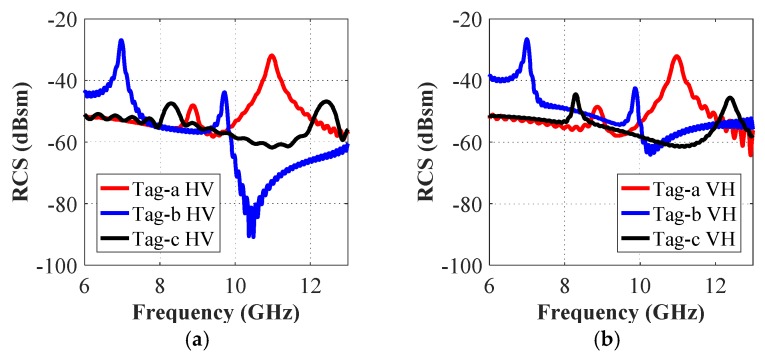
Cross-polar RCS simulation results of tag-a, -b, and -c: (**a**) Cross-polar HV RCS responses; (**b**) Cross-polar VH RCS responses.

**Figure 14 sensors-19-04785-f014:**
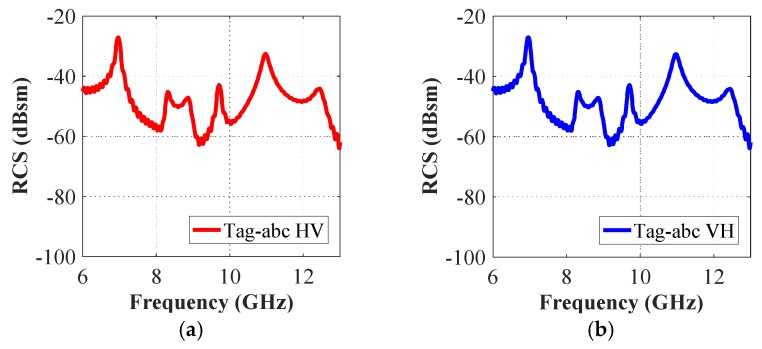
Cross-polar RCS simulation results of tag-abc: (**a**) Cross-polar HV RCS responses; (**b**) Cross-polar VH RCS responses.

**Figure 15 sensors-19-04785-f015:**
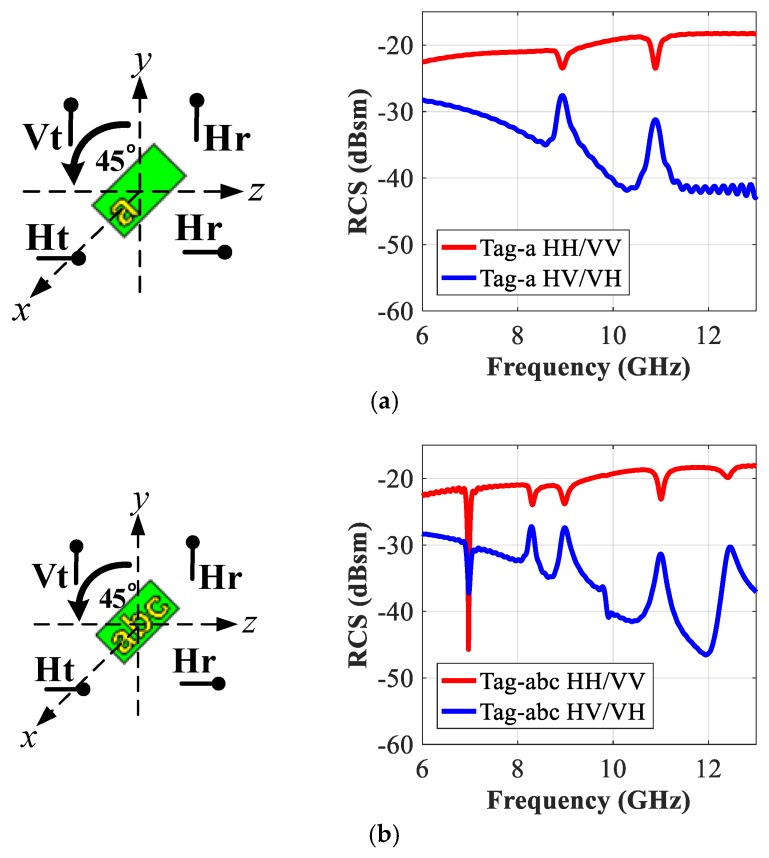
RCS simulation results of tag oriented to 45° in both co- and cross-polarizations: (**a**) Tag-a; (**b**) Tag-abc.

**Figure 16 sensors-19-04785-f016:**
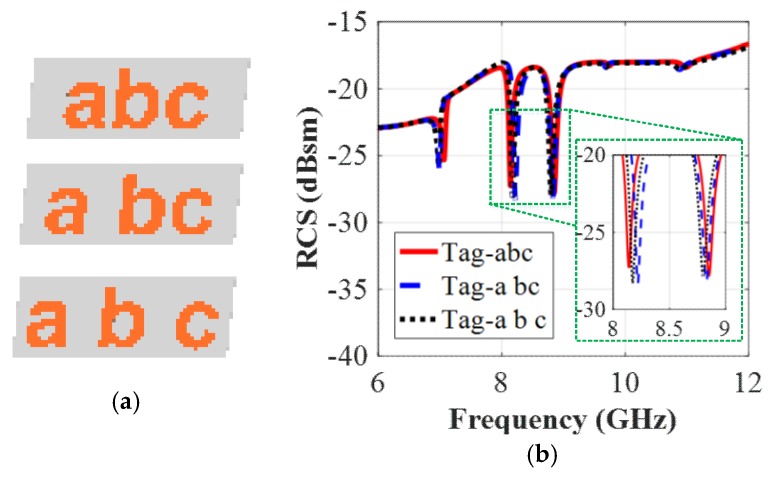
RCS simulation results of the tag-abc with and without space between the letters: (**a**) Different realized tags with and without click space; (**b**) Co-polar HH RCS responses.

**Figure 17 sensors-19-04785-f017:**
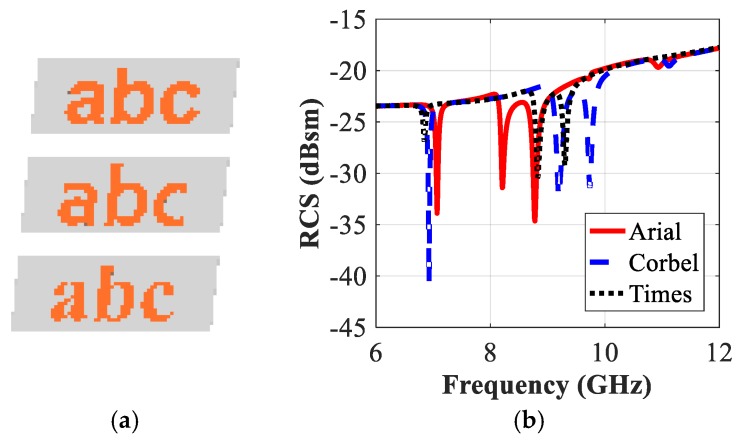
RCS Simulation Results of tag-abc printed by various fonts: (**a**) Different tags written by Arial, Corbel and Time New Roman fonts; (**b**) Co-polar HH RCS responses.

**Figure 18 sensors-19-04785-f018:**
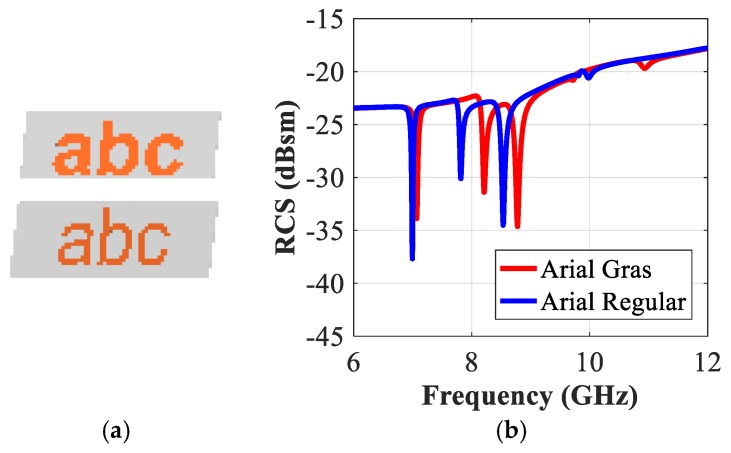
RCS Simulation results of tag-abc printed by Arial font with different styles: (**a**) Tag-abc written by Arial gars and regular style; (**b**) Co-polar HH RCS responses.

**Figure 19 sensors-19-04785-f019:**
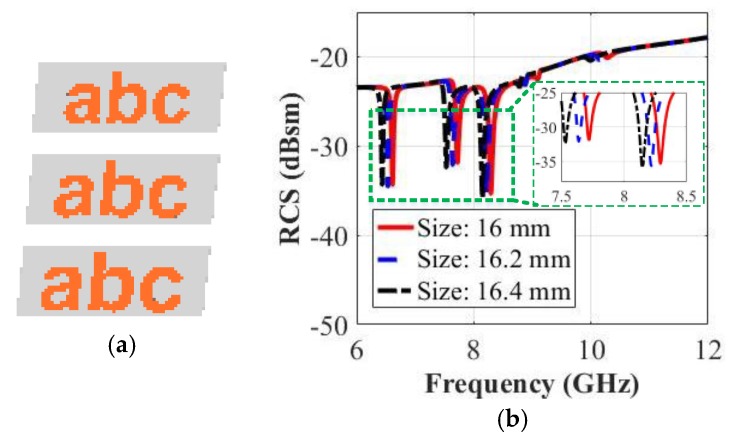
RCS simulation results of tag-abc written by different heights: (**a**) Different tags written by 16 mm, 16.2 mm and 16.4 mm sizes; (**b**) Co-polar HH RCS responses.

**Figure 20 sensors-19-04785-f020:**
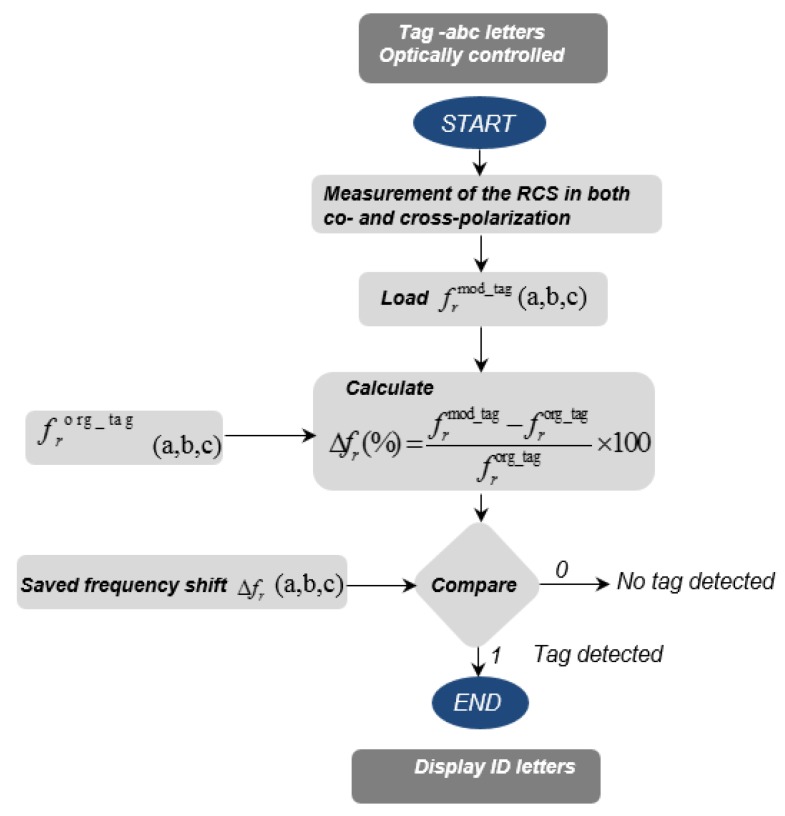
Algorithmic flow chart of the proposed method for robust detection.

**Figure 21 sensors-19-04785-f021:**
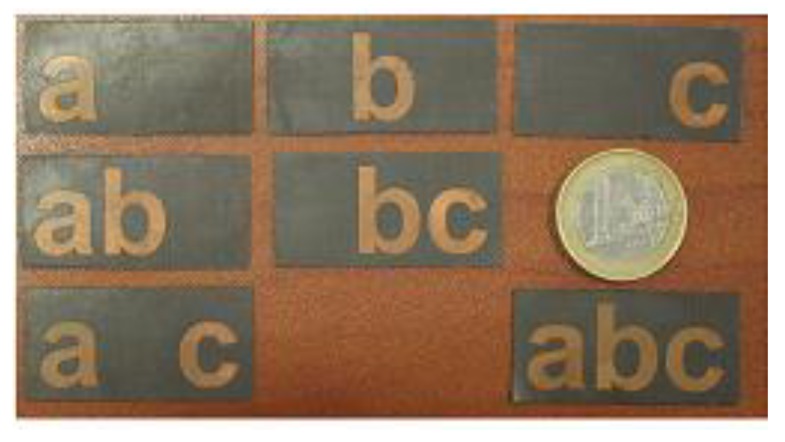
Set of manufactured letter-IDs tags.

**Figure 22 sensors-19-04785-f022:**
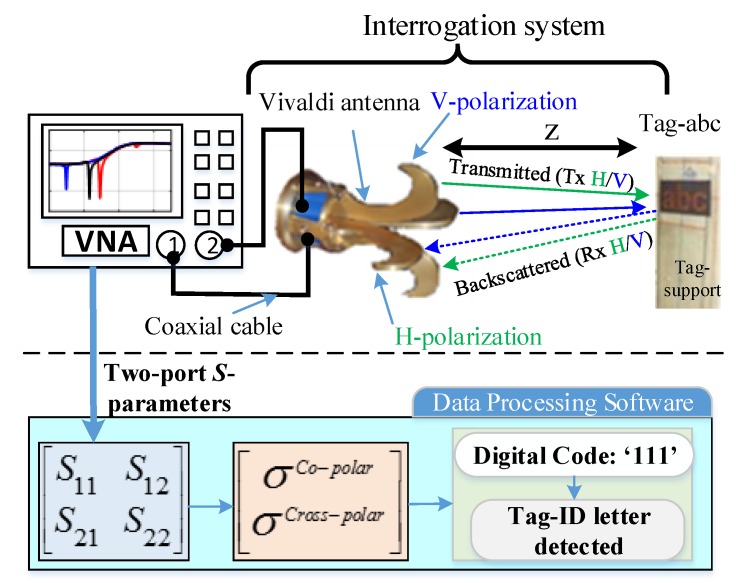
Strategy of measurement for the identification of chipless tags.

**Figure 23 sensors-19-04785-f023:**
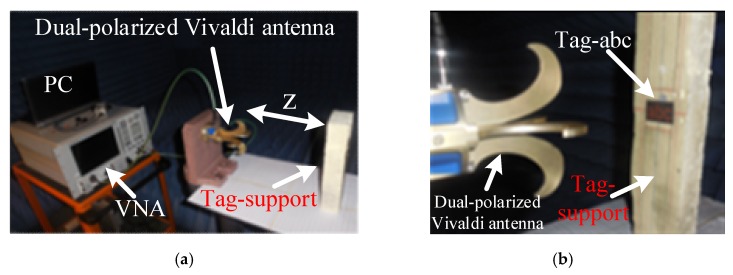
Photographs of measurement in anechoic chamber: (**a**) Measurement setup; (**b**) Tag-abc under measurements.

**Figure 24 sensors-19-04785-f024:**
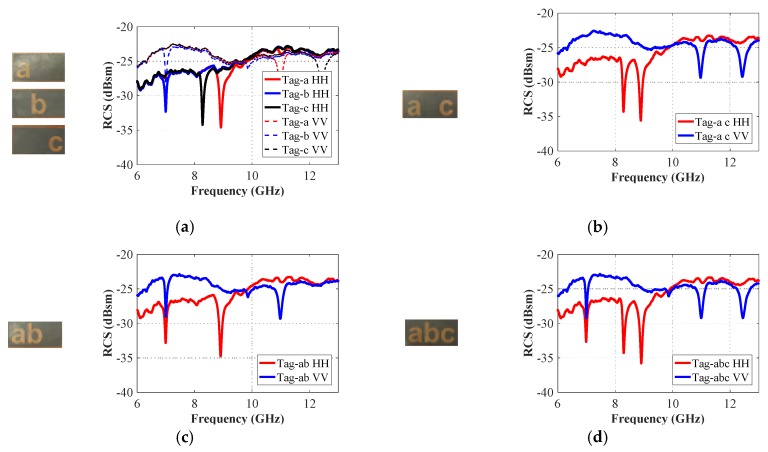
Measurement co-polar (HH/VV) RCS results: (**a**) Tag-a, -b, and -c; (**b**) Tag-a c; (**c**) Tag-ab; (**d**) Tag-abc.

**Figure 25 sensors-19-04785-f025:**
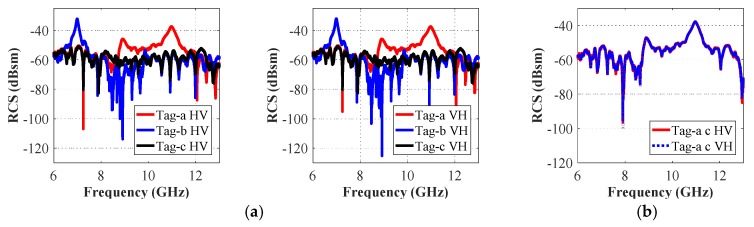
Measurement cross-polar (HV/VH) RCS results: (**a**) Tag-a, -b, and -c; (**b**) Tag-a; (**c**) Tag-ab; (**d**) Tag-abc.

**Table 1 sensors-19-04785-t001:** Unique frequency signature of single tag letters from Figure.

Tag-ID	Resonant Frequency/Polarization	
	$f_{HH}\;(\mathrm{GHz})$	$f_{VV}\;(\mathrm{GHz})$
Tag-a	8.90	10.97
Tag-b	6.96	6.97
Tag-c	8.32	12.44

**Table 2 sensors-19-04785-t002:** Summary of the frequency shifts in co-polar-HH achieved with the simulated parameters of designed tags (frequency shift is expressed in %).

Freq. Shift (%)	Font Name (Height = 15 mm)			Height (mm)		Spacing	
	Arial Reg.	Time	Corbel	16	16.4	a b c	a b c
$\Delta f_{r}^{}$ (‘a’)	−4.15	4.49	9.32	−6.85	−8.53	−0.89	0.78
$\Delta f_{r}^{}$ (‘b’)	0.43	−1.72	−0.43	−5.02	−7.61	0.57	0.14
$\Delta f_{r}^{}$ (‘c’)	−6.12	6.00	10.45	−7.21	−9.49	−1.20	−1.80

**Table 3 sensors-19-04785-t003:** Set of different values of three geometrics parameters of each resonator letter.

Font	Regular	F_1	F_2	F_3	F_4
	Bold	F_1	F_2	F_3	F_4
Size		Size_1	Size_2	Size_3	Size_4
Spacing		abc	a bc	ab c	a b c

**Table 4 sensors-19-04785-t004:** Comparison of proposed design and robust coding technique with RFID chipless printable alphabets legacy designs.

Design Reference	Design Type	Inclusion of Tag Bottom Ground Plane	Font Type (Height in mm)	Frequency (GHz)	No. of Resonators Per Tag	Size (cm^2^)	Coding Spatial Density (bit/cm^2^)	Optically Controlled Resonators	Reconfigurable	Analysis of Structural (Font Types, Font Size, Orientation, Inter-Letter Spacing, etc.) Modifications
[25]	Mender line dipole structure with attached text	No	-	0.896	1	2.5 × 13.5	-	Yes	No	No
[14]	Single letter Latin Alphabet	No	Arial (24)	1–10	1	3.7 × 3.7	0.34	Yes	No	No
[27]	Single letter Latin Alphabet	No	Arial (48)	1–10	1	4.8 × 4.5	0.21	Yes	No	No
[2]	Single letter Arabic Alphabets	No	Arial (24)	1–10	1	3.7 × 3.7	0.35	Yes	No	No
[1]	Peyote symbols of five letters	No	Blocks building	57–64	5	5 × 1	2.00	Yes	Yes	No
[21]	Latin alphabets letters and words with opening slots	No	Calibri (53)	1–10	3	5.3 × 9	0.06	Yes	No	No
This work	Latin Alphabet letters and three-letter words	Yes	Arial, Corbel, Times New Roman (different sizes)	6–13	3	4 × 2	2.63 **	Yes	Yes	Yes

** Minimum coding density.

**Table 5 sensors-19-04785-t005:** Tag-ID Letters’ digital code and resonant frequency signatures (GHz) (ø represents no letter).

Tag-ID Letters	Digital Code	Signature Frequency/Polarization		
		*f* _HH_	*f* _VV_	*f*_VH_ = *f*_HV_
Tag-aøø	100	8.91	10.98	8.90, 10.97
Tag-øbø	010	7.00	6.99	7.00, 6.99
Tag-øøc	001	8.27	12.41	8.27, 12.41
Tag-abø	110	7.00 8.91	6.99 10.98	7.00, 8.91, 6.99, 10.98
Tag-aøc	101	8.27 8.97	10.98 12.41	8.27, 8.97, 10.98, 12.41
Tag-øbc	011	7.00 8.27	6.99 12.41	7.00, 8.27, 6.99, 12.41
Tag-abc	111	8.91 7.00 8.27	10.98 6.99 12.41	8.81, 7.00, 8.27 10.98, 6.99, 12.41

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
