# Peer review of "Robust Detection for Chipless RFID Tags Based on Compact Printable Alphabets"

_sensors, 2019, doi:10.3390/s19214785_

Round 1
Reviewer 1 Report
This article presents a way of printing RFID tags with a visual design. The idea is that the RFID tags can be an integral part of a printed logo. The results of experiments are shown in which the shape of the tag is clearly detectable through their radar cross section. Experiments were conducted with different letter orientations and different fonts. Both simulations and practical experiments were conducted. The results are interesting and encouraging. (However, a limitation of the study is that only three distinct letters were used in the experiments.)
An interesting "future work" item would be to determine the relationship, if any, between the ability to recognise tags and visually see letters. As the authors point out on Line 196, the letters "i" and "l" are hard to distinguish in both cases. It would also be interesting to know if a special-purpose typeface can be designed for this RFID application which would be easier to recognise, like the old OCR and MICR fonts.
The paper is refreshingly well written and easy to read. A few small corrections and deletions, in angled brackets:
Lines 78-79: "complex resonant structures as proposed <previously> [4, 6-9, 12, 13, 19, 22, 24]"
Line 98: "be pointed out that <Boularess et al.'s> study [2] has"
Lines 98-99: "while <Singh et al.'s work> [14, 27]"
Line 99: "It has been shown <by Demir et al.> [21] that"
Line 110: "from <> approaches< described previously> [1, 2, 14, 21, 25, 27]"
Line 117: "as proposed <elsewhere> [4, 6-9, 12, 13, 19"
Line 166: "words<.>"
Line 283: "when compared to <previously> proposed designs <> [1, 21]
Line 359: Inconsistent quotation marks
Line 442: "In <Equation> (1)"
Line 451: "results of <> Figs."
Line 452: "by using <Equation> (1)"
Line 508: "tag reported <previously> [4, 6-9, 12, 13, 19, 22, 24]"
Line 513: "designs previously reported <> [2, 14, 21, 25, 27]."
Line 514: "in <a> recently reported study"
Line 517: "to those described <elsewhere> [2, 14, 21, 25, 27]"
Line 540: "A calibration technique mentioned <by Hotte et al.> [30] is used"
Line 552: "from <Equations> (5) and (6)"
Line 557: "In <Equations> (5) and (6)"
Line 570: "by using <Equations> (5) and (6)"
Line 581: "by using <Equation> (6)"
Author Response
Thank you very much for your taking time to review our work and your nice suggestions to improve the quality of our manuscript. As per your advice the suggested items about the determination of the relationship between the recognized tags and visually identified tags has been added in the revised manuscript as the future work. Also, the analysis of the designing of the special kind of typeface for such RFID tags for their easier recognition could be very interesting future work and our group is strongly considering this suggestion for the possible future work. The findings of these future work could be reported in near future. The revised manuscript has been updated with these potential future research items as the last paragraph of section 7.4 on page 23.
Reviewer 2 Report
I suggest that the advantages of this approach over the approach of using optical means of identification should be well highlighted in this paper. Optical systems can read alphabets the alphabetic materials over longer distance and these optical systems are not affected by electromagnetic noise. The authors should clearly shown the advantage of the RF method over optical methods. One advantage of the system presented by the authors is the possibility of reading alphabetic materials in the dark. This is something optical systems can not do.
Furthermore, the authors should present how well their detection system will fare in the presence of external electromagnetic noise.
Author Response
Than
Thank you for your nice question and pointing out towards an important direction of effect of noise on detection process.
Firstly, as pointed out by you, optical system could be used for the tags reading over the long distances. However, the optical system requires direct contact with the object to be identified. On the other hand, the RF system can even locate the objects at the sufficient long distances and even in the dark, which makes this method a favorable choice for the different smart applications such as in IOT and cost-effective inventory management, etc.
The presence of external EM noise could bring a change in the resonance frequency and depth of the resonance value of the frequency signature of the tag-ID. For such cases different post data post processing techniques like quantile regression models [1] and random sample consensus (RANSAC) algorithms [2] with other approaches such as anti-collision time domain techniques [3] and multi-antenna detection system [4] could be used to improve the detection efficiency and capabilities of the proposed robust RF-based detection system with the mitigation and suppression of the coupling noise.
The summary of the above answers is also added in the revised manuscript at the end of the Section 7.4 on page 22-23 of the revised manuscript (highlighted in red).
Thank you very much for your taking time to review our work and your nice suggestions to improve the quality of our manuscript. We hope the updated manuscript satisfy your requirements.
[1] M. Manekiya, M. Donelli, A. Kumar, and K. S. Menon, "A Novel Detection Technique for a Chipless RFID System Using Quantile Regression," Electronics, vol. 7, no. 12, 2018.
[2] C. Jing et al., "A Robust Noise Mitigation Method for the Mobile RFID Location in Built Environment," Sensors, vol. 19, no. 9, 2019.
[3] G. Khadka and S.-S. Hwang, "Tag-to-Tag Interference Suppression Technique Based on Time Division for RFID," (in eng), Sensors (Basel, Switzerland), vol. 17, no. 1, p. 78, 2017.
[4] J. D. Griffin and G. D. Durgin, "Gains For RF Tags Using Multiple Antennas," IEEE Transactions on Antennas and Propagation, vol. 56, no. 2, pp. 563-570, 2008.

Reviewer 3 Report
It is good novel work. Please give some more samples such as “aaa”,"aab" etc. Please compare "a","aa" and "aaa".
Author Response
Thank you very much for your taking time to review our work and your nice suggestions to improve the quality of our manuscript.
As per your advice, the comparison of the results of more samples “a”, “aa”, “aaa”, “aab”, “abb”, “bba” has been added in the revised manuscript as Fig. 10. The results and discussion about these newly added samples results has been given in Section 4.1 on page 9 of the revised manuscript.
